# Jets in the Magnetosheath: IMF Control of Where They Occur

Laura Vuorinen[1], Heli Hietala[1,2], and Ferdinand Plaschke[3]

[1]Department of Physics and Astronomy, University of Turku, Turku, Finland
[2]Department of Earth, Planetary, and Space Sciences, University of California, Los Angeles, USA
[3]Space Research Institute, Austrian Academy of Sciences, Graz, Austria

**Correspondence:** L. Vuorinen (lakavu@utu.fi)

**Abstract.** Magnetosheath jets are localized regions of plasma that move faster towards the Earth than the surrounding magnetosheath plasma. Due to their high velocities, they can cause indentations when colliding into the magnetopause and trigger processes such as magnetic reconnection and magnetopause surface waves. We statistically study the occurrence of these jets in the subsolar magnetosheath using measurements from the five Time History of Events and Macroscale Interactions during Substorms (THEMIS) spacecraft and OMNI solar wind data from 2008–2011. We present the observations in the $\mathbf{B}_{\text{IMF}}$-$\mathbf{v}_{\text{SW}}$-plane and study the spatial distribution of jets during different interplanetary magnetic field (IMF) orientations. Jets occur downstream of the quasi-parallel bow shock approximately 9 times as often as downstream of the quasi-perpendicular shock, suggesting that foreshock processes are responsible for most jets. For oblique IMF, with 30°–60° cone angle, the occurrence increases monotonically from the quasi-perpendicular side to the quasi-parallel side. This study offers predictability for the numbers, locations, and magnetopause impact rates of jets observed during different IMF orientations allowing us to better forecast the formation of these jets and their impact on the magnetosphere.

## 1 Introduction

The varying solar wind and interplanetary magnetic field (IMF) conditions contribute to the dynamic nature of the Earth's magnetosphere. The orientation of the IMF determines the location of the turbulent foreshock region, formed by the interaction of the inflowing solar wind with particles reflected from the shock (e.g., Eastwood et al., 2015). In addition to the global scale structure of the magnetosphere, there are various types of local spatial and temporal variations caused either by discontinuities in the solar wind or by the non-linear evolution of the system itself. Some examples of local variations include foreshock transients (e.g., Schwartz and Burgess, 1991), magnetopause surface waves (e.g., Plaschke et al., 2009) and transient structures in the magnetosheath (e.g., Plaschke et al., 2018).

One and the most common type of magnetosheath transients are local dynamic pressure enhancements called magnetosheath jets (Plaschke et al., 2018, and the references therein). These are plasma regions that exhibit higher earthward dynamic pressure than the surrounding magnetosheath plasma due to high earthward velocities (Plaschke et al., 2013). A typical size of these

jets perpendicular to their flow direction is around 1 $R_{\mathrm{E}}$ and jets larger than 2 $R_{\mathrm{E}}$ in diameter can be considered geoeffective (Plaschke et al., 2016). If these jets hit the magnetopause, they can indent the magnetopause, produce magnetopause waves and trigger phenomena that may also affect the inner magnetosphere. For example, Hietala et al. (2018) published observational evidence of a jet triggering magnetic reconnection at the dayside magnetopause. Wang et al. (2018) showed direct correspondence between magnetosheath jets and diffuse and discrete auroral brightenings. The newly observed magnetopause surface eigenmodes (standing waves) were also excited by a jet colliding into the magnetopause (Archer et al., 2019). It is fair to say that magnetosheath jets play a role in energy transport in the Earth's magnetosphere.

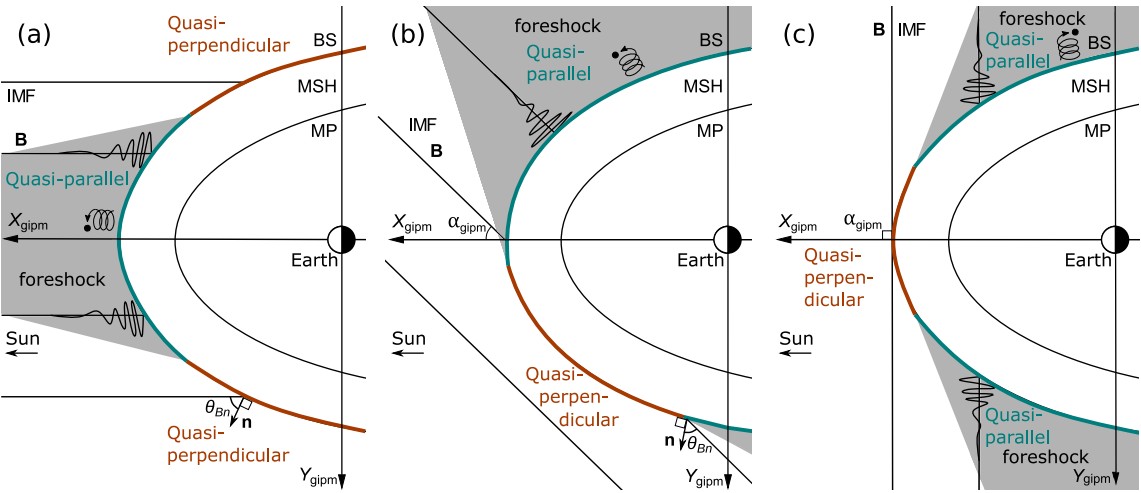

**Figure 1.** A sketch of the approximate foreshock regions (filled with gray) and the quasi-parallel (turquoise) and the quasi-perpendicular (brown) parts of the bow shock (BS) for IMF cone angles: (**a**) $\alpha_{\mathrm{gipm}} \sim 0°$ (radial IMF) (**b**) $\alpha_{\mathrm{gipm}} \sim 45°$ (e.g., Parker spiral IMF), and (**c**) $\alpha_{\mathrm{gipm}} \sim 90°$. These are presented in the plane containing the solar wind velocity vector (anti-parallel to the $X_{\mathrm{gipm}}$-axis) and the IMF. The magnetopause (MP) and the magnetosheath (MSH) are shown downstream of the bow shock.

How these jets are formed is still an open question, although many different models have been suggested (Plaschke et al., 2018). Hietala et al. (2009) proposed a jet formation mechanism in which jets are generated by local curvature variations of the bow shock, called bow shock ripples. According to the Rankine-Hugoniot jump conditions, the upstream plasma velocity component parallel to the local shock normal is decelerated most efficiently. This means that if the flow is not parallel to the shock normal, e.g., if the shock surface is locally inclined, the plasma flow will be compressed but less decelerated. A shock ripple could therefore produce a magnetosheath jet. Large scale rippling is thought to be more prevalent in the quasi-parallel region of the bow shock where the angle between the IMF and the bow shock normal is small ($\theta_{Bn} < 45°$). Therefore, based on the above, we would expect more jets downstream of the quasi-parallel bow shock sections. The locations of the quasi-parallel areas and the foreshock regions change with the IMF orientation as illustrated in Fig. 1 for three different IMF orientations.

Short large amplitude magnetic structures (SLAMS) (Schwartz and Burgess, 1991) in the foreshock advecting towards the bow shock have also been proposed to affect jet generation in two alternative ways (Plaschke et al., 2018). First, by forming

bow shock ripples by merging into the bow shock and thereby producing jets by the ripple mechanism explained above. Second, Karlsson et al. (2015) suggested that SLAMS could transform into jets when travelling through a dent in the bow shock. Recently, Palmroth et al. (2018) ran a global hybrid-Vlasov simulation to study magnetosheath jets and the jet under scrutiny appeared to be a SLAMS-like structure going through the shock.

While jets are mostly observed during steady IMF, a minority of jets can be explained by solar wind discontinuities, specifically by sharp variations in IMF orientation (Archer and Horbury, 2013). Jets associated with solar wind discontinuities are not linked to the quasi-parallel bow shock only. It has been suggested by Archer et al. (2012) that jets could form when the shock locally changes from quasi-parallel to quasi-perpendicular or vice versa as an IMF discontinuity passes by.

     Previous studies (Plaschke et al., 2013; Archer and Horbury, 2013) have shown that the only variable strongly controlling

the occurrence of local dynamic pressure enhancements in the subsolar magnetosheath is the IMF cone angle between the IMF and the Earth-Sun line. According to these studies, such transients occur predominantly during low IMF cone angle conditions, that is when the angle $\alpha$ between the IMF and the Earth-Sun line is less than $45°$. This result supports the predictions of the ripple and SLAMS models because the quasi-parallel region is mostly upstream of the subsolar magnetosheath during low cone angle IMF. Furthermore, Archer and Horbury (2013) have specifically shown that the occurrence rate of dynamic

pressure enhancements is higher downstream of the quasi-parallel part of the bow shock supporting the formation mechanisms associated with the quasi-parallel shock. However, the definitions of the local dynamic pressure enhancements are different in these two studies. Archer and Horbury (2013) (from here on AH13) defined their dynamic pressure threshold by the background magnetosheath dynamic pressure. Plaschke et al. (2013) (from here on P13) set their threshold based on the solar wind dynamic pressure, and specifically defined jets as enhancements of anti-sunward dynamic pressure to study transients that

could potentially hit the magnetopause and have effects on the magnetosphere. In addition, the observations used in the studies were from different years: 2008 (AH13) and 2008–2011 (P13).

     To study the overlap between these two definitions, both selection criteria were recently applied to the data of AH13 study in the subsolar magnetosheath of $30°$ solar zenith angle (Plaschke et al., 2018). 17 % of the events corresponding to the criteria of AH13 also corresponded to the criteria of magnetosheath jets by P13, and they made up 47 % of all these jets. This means

that 83 % of the AH13 events were not magnetosheath jets defined by P13 and, on the other hand, 53 % of the P13 events were not in the AH13 set. For example, flux transfer events (FTEs) close to the magnetopause are included for the selection criteria of AH13 but not when the P13 selection criteria are applied. Thus, there are significant disparities between these two types of plasma entities and therefore the result of AH13 cannot be straightforwardly generalized to jets by P13. Furthermore, AH13 estimated the angle $\theta_{Bn}$ between the shock normal and the IMF with a magnetosheath streamline model. Such a method of

tracing streamlines back to the shock is not suitable for magnetosheath jets defined by P13 because the median deflections of jets from the background flow are between $20°$–$45°$ (Hietala and Plaschke, 2013).

     In this paper, we investigate for the first time how the spatial occurrence and the magnetopause impact rates of jets in the subsolar magnetosheath studied by Plaschke et al. (2013) relate to the IMF orientation. We use data gathered during the years 2008–2011 from the five Time History of Events and Macroscale Interactions during Substorms (THEMIS) spacecraft in the

magnetosheath (Angelopoulos, 2008) and NASA OMNI high resolution solar wind data (King and Papitashvili, 2005). We

compare the locations of jet observations to the location of the quasi-parallel bow shock to test the validity of jet formation mechanisms based on the nature of the quasi-parallel bow shock and to provide quantitative predictions of their occurrence rates.

## 2 Data and Methods

### 2.1 Observational Data Sets

The data set used in this study and the jet selection criteria are described in detail by Plaschke et al. (2013). Here we briefly explain the key steps of selecting the magnetosheath and jet observations. The magnetosheath (MSH) data were selected from the measurements taken by the five THEMIS spacecraft during the years 2008–2011 within a $30°$ wide Sun-centered cone with its tip at Earth and within 7–18 Earth radii from the center of Earth. The observations made by different instruments were interpolated to a common timeline at 1 second cadence. Magnetosheath observations were selected by requiring the density to be over twice the solar wind density and the energy flux of 1 keV ions to be larger than the flux of 10 keV ions to ensure that the spacecraft were not inside the magnetosphere. The solar wind (SW) conditions were calculated as averages of the OMNI measurements from the preceding five minutes. These criteria yield 2,736.9 h of magnetosheath and solar wind data.

The main criterion for a magnetosheath jet is to have dynamic pressure ($P_{\mathrm{dyn}} = \rho v^2$) in the anti-sunward $X_{\mathrm{GSE}}$-direction that exceeds half of the SW dynamic pressure:

$$P_{\mathrm{dyn,MSH},X} = \rho_{\mathrm{MSH}} v_{\mathrm{MSH},X}^2 > \frac{1}{2} P_{\mathrm{dyn,SW}} = \frac{1}{2} \rho_{\mathrm{SW}} v_{\mathrm{SW}}^2. \tag{1}$$

The entire jet interval is then defined as the period when the earthward dynamic pressure is over $1/4$ of the total solar wind dynamic pressure. The moment of the highest ratio between the MSH and SW dynamic pressures within the jet interval is denoted as $t_0$ and the data set of jet observations consists of the measurements taken at these times. In order to prevent multiple consecutive peaks from being counted as individual jets, within the one-minute long windows before and after the jet interval, the $X_{\mathrm{GSE}}$ ion velocity has to go above half of the corresponding value at $t_0$. The data set contains 2,859 jets.

### 2.2 Coordinate System

In order to compare the positions of jets to the location of the quasi-parallel shock during different solar wind and IMF conditions, we first need to move to the plane containing the IMF. We use the geocentric interplanetary medium reference frame (GIPM) introduced by Bieber and Stone (1979). The GIPM reference frame has been used in many magnetosheath studies, e.g., by Verigin et al. (2006) and Dimmock and Nykyri (2013). The coordinate system is visualized in Fig. 2. In this reference frame, the $X_{\mathrm{gipm}}$-axis is anti-parallel to the solar wind velocity vector $\mathbf{v}_{\mathrm{SW}}$ while also taking into account the orbital aberration caused by Earth's orbital speed of $\sim 30$ km/s. The $Y_{\mathrm{gipm}}$-axis is defined in the plane containing the IMF and the $X_{\mathrm{gipm}}$-axis (the $\mathbf{B}_{\mathrm{IMF}}$-$\mathbf{v}_{\mathrm{SW}}$-plane). The unit vectors of the GIPM reference frame as functions of GSE vectors $\mathbf{v}_{\mathrm{SW}} = (v_X, v_Y, v_Z)$ and

$\mathbf{B}_{\mathrm{IMF}} = \mathbf{B} = (B_X, B_Y, B_Z)$ are given by (Verigin et al., 2006):

$$\hat{\mathbf{X}}_{\mathrm{gipm}} = \frac{(-v_X, -v_Y - 30\ \mathrm{km/s}, -v_Z)}{\sqrt{v_X^2 + (v_Y + 30\ \mathrm{km/s})^2 + v_Z^2}} \tag{2}$$

$$\hat{\mathbf{Y}}_{\mathrm{gipm}} = \begin{cases} \dfrac{(-\mathbf{B} + (\mathbf{B} \cdot \hat{\mathbf{X}}_{\mathrm{gipm}})\hat{\mathbf{X}}_{\mathrm{gipm}})}{|\mathbf{B} - (\mathbf{B} \cdot \hat{\mathbf{X}}_{\mathrm{gipm}})\hat{\mathbf{X}}_{\mathrm{gipm}}|}, & \text{if } \mathbf{B} \cdot \hat{\mathbf{X}}_{\mathrm{gipm}} > 0 \\ \dfrac{(\mathbf{B} - (\mathbf{B} \cdot \hat{\mathbf{X}}_{\mathrm{gipm}})\hat{\mathbf{X}}_{\mathrm{gipm}})}{|\mathbf{B} - (\mathbf{B} \cdot \hat{\mathbf{X}}_{\mathrm{gipm}})\hat{\mathbf{X}}_{\mathrm{gipm}}|}, & \text{if } \mathbf{B} \cdot \hat{\mathbf{X}}_{\mathrm{gipm}} < 0 \end{cases} \tag{3}$$

$$\hat{\mathbf{Z}}_{\mathrm{gipm}} = \hat{\mathbf{X}}_{\mathrm{gipm}} \times \hat{\mathbf{Y}}_{\mathrm{gipm}}. \tag{4}$$

5   In this coordinate system the IMF cone angle

$$\alpha_{\mathrm{gipm}} = \arccos\left(|\mathbf{B} \cdot \hat{\mathbf{X}}_{\mathrm{gipm}}|/B\right) \in [0°, 90°] \tag{5}$$

is always in the quadrant of the $X_{\mathrm{gipm}}$-$Y_{\mathrm{gipm}}$-plane where $X_{\mathrm{gipm}} > 0$ and $Y_{\mathrm{gipm}} < 0$. This means that the quasi-parallel region of the bow shock is mostly on the negative side of the $Y_{\mathrm{gipm}}$-axis (Fig. 2). Investigating $Y_{\mathrm{gipm}}$ allows us to compare the observations made downstream of the quasi-parallel and quasi-perpendicular bow shock regions.

10   As illustrated in Fig. 1, the location of the quasi-parallel region varies for different IMF cone angle conditions. Therefore, we divide the data set into three cone angle ranges for comparison: quasi-radial IMF ($\alpha_{\mathrm{gipm}} \in [0°, 30°)$) when almost all of the dayside magnetosheath observations can be considered to be downstream of the quasi-parallel shock, oblique IMF ($\alpha_{\mathrm{gipm}} \in [30°, 60°)$), and high cone angle IMF ($\alpha_{\mathrm{gipm}} \in [60°, 90°]$) when almost all of the dayside magnetosheath observations can be considered to be downstream of the quasi-perpendicular shock. The number of jets and the median value of $\alpha_{\mathrm{gipm}}$ in 15   each range are respectively: 970 & $21.4°$, 1,403 & $47.3°$ and 486 & $75.2°$. These limits were chosen such that each of the ranges has representable numbers of observations, and because the locations of the expected quasi-parallel regions are clearly different in these three ranges.

## 2.3  Normalization of Positions by the Solar Wind Dynamic Pressure

The size of the magnetosphere-bow shock system changes slightly during different solar wind conditions. To account for these 20   changes and to make observations directly comparable to each other during varying conditions, we normalize all spacecraft positions (subscript 0) to the mean solar wind dynamic pressure of all observations assuming protons only: $\overline{P}_{\mathrm{dyn,SW}} = 1.76$ nPa. The normalization acts only on the distance from Earth, not the direction, and is calculated with the commonly used relation (e.g., Merka et al., 2005; Spreiter et al., 1966):

$$r_{\mathrm{n}} = r_0 \left( \frac{P_{\mathrm{dyn,SW,0}}}{\overline{P}_{\mathrm{dyn,SW}}} \right)^{1/6}. \tag{6}$$

25  ## 2.4  Renormalization by All Magnetosheath Observations

We bin the observations as a function of $Y_{\mathrm{gipm}}$, constructing histograms of the jet occurrence rates. The distributions of jets are renormalized by the distributions of all magnetosheath observations to account for the time spent under different conditions.

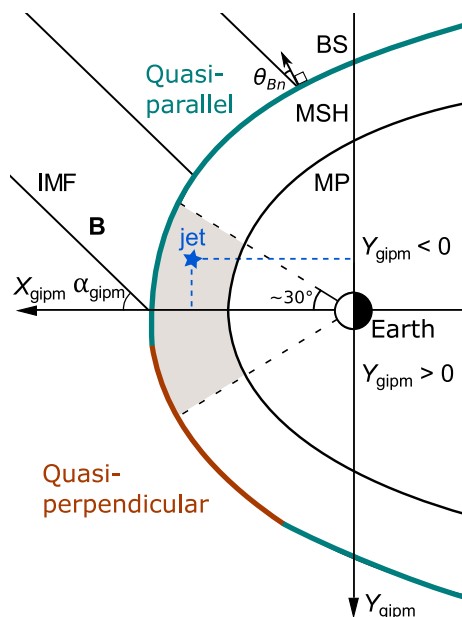

**Figure 2.** The GIPM reference frame has the $X_{\mathrm{gipm}}$-axis anti-parallel to the solar wind velocity vector and the $Y_{\mathrm{gipm}}$-axis perpendicular to it in the plane containing the IMF. The $Y_{\mathrm{gipm}}$-axis always points to the more quasi-perpendicular side. The grey area represents the observation area in the subsolar magnetosheath. In the GSE frame, this is a $30°$ cone around the Earth-Sun line. The star is an example of a jet observation at $t_0$.

The normalized occurrence rates are presented in the units of jets per unit time. The histogram error bars are 95% binomial proportion confidence intervals calculated using the conservative Clopper-Pearson method (e.g., Brown et al., 2001).

## 2.5  2D Maps

We plot 2D maps of the jet occurrence in the $X_{\mathrm{gipm}}$-$Y_{\mathrm{gipm}}$-plane. Similar to the histograms, the positions are normalized by the mean dynamic pressure 1.76 nPa using Eq. (6), and then the jet distributions are renormalized by the MSH observations. We set the lower limit of MSH observations to 1,000 per cell because that removes the cells with very high uncertainties. We plot model bow shocks and magnetopauses to aid visualization using models by Merka et al. (2005) and Shue et al. (1998), respectively. The models have been calculated for each cone angle range separately but for $\overline{P}_{\mathrm{dyn,SW}} = 1.76$ nPa. The bow shock model depends on the Alfvén Mach number whose values for each cone angle range are: $M_A = 11.5$, $M_A = 9.92$ and $M_A = 9.74$ (from the lowest to the highest cone angle range). The model magnetopauses have been calculated using the median values of IMF $B_{Z,\mathrm{GSM}}$-components (0.066 nT, $-0.143$ nT and 0.332 nT, from the lowest to the highest cone angle range) as parameters. The original model is symmetric around the $X$-axis of the aberrated GSE coordinate system that includes the correction for the Earth's orbital motion. We have used the $X_{\mathrm{gipm}}$-axis (anti-parallel to $\mathbf{v}_{\mathrm{SW}}$) as the axis of symmetry.

## 2.6 Estimating the Magnetopause Impact Rates

To estimate the number of jets impacting on the magnetopause, we use the model published by Plaschke et al. (2016). It is a statistical model created using the same data set as in this study, and it is based on the distribution of jet diameters $D_\perp$ perpendicular to their propagation direction. This probability distribution of perpendicular sizes was calculated using 662 multispacecraft jet observations, of which 655 were made by the inner THEMIS A, D and E spacecraft. Therefore, we only use THA, THD and THE data for the estimation of impact rates. According to the model, the impact rate $Q_{\mathrm{imp}}$ of jets larger than $D_{\perp\mathrm{min}}$ per unit time is:

$$Q_{\mathrm{imp}} = \frac{4 A_{\mathrm{ref}} \cos\phi \, Q_{\mathrm{obs}}}{\pi D_{\perp 0}} \int\limits_{D_{\perp\mathrm{min}}}^{\infty} e^{-D_\perp / D_{\perp 0}} \frac{\mathrm{d}D_\perp}{D_\perp^2}, \tag{7}$$

where $Q_{\mathrm{obs}}$ is the observed rate of jet occurrence per unit time, $\phi$ is the mean angle of jet propagation with respect to the $-\hat{\mathbf{X}}_{\mathrm{GSE}}$ unit vector, $D_{\perp 0} = 1.34\,R_{\mathrm{E}}$ is a model parameter determined from the observations, and $A_{\mathrm{ref}} = 102\,R_{\mathrm{E}}^2$ is the circular reference area of the $30°$ solar zenith angle subsolar magnetopause that we are estimating the impact rates for. The jet occurrence rates $Q_{\mathrm{obs}}$ and mean propagation angles $\phi$ for the three different cone angle ranges based on THA, THD, and THE observations are: $2.90\,\mathrm{h}^{-1}$ & $25.9°$, $1.32\,\mathrm{h}^{-1}$ & $25.1°$, and $0.267\,\mathrm{h}^{-1}$ & $23.7°$ (from the lowest to the highest cone angle range).

## 3 Results

In Figure 3 we present histograms of the number of jets detected per hour per $R_{\mathrm{E}}$ in the magnetosheath as functions of $Y_{\mathrm{gipm}}$ for each cone angle range. The histograms are cropped to $Y_{\mathrm{gipm}} \in [-8, 8]$ to avoid the outermost bins with very large error bars. This leaves out 2 jets and 3,875 MSH observations in total.

For the highest IMF cone angle values $[60°, 90°]$ when the subsolar magnetosheath is mostly downstream of the quasi-perpendicular shock, the distribution is flat but has higher bins around the edges, though within the error bars. A typical occurrence rate here is around one jet in five hours per $R_{\mathrm{E}}$. In comparison, for cone angle ranges $[0°, 30°)$ and $[30°, 60°)$ corresponding to quasi-radial and oblique IMF, the occurrence of jets is higher on the negative part of the $Y_{\mathrm{gipm}}$-axis. In particular, for oblique IMF cone angles $[30°, 60°)$ there is a clear increasing trend with decreasing $Y_{\mathrm{gipm}}$, i.e., towards the side of the shock which is generally more quasi-parallel. We can see that the occurrence rates during oblique IMF almost coincide with high cone angle IMF occurrence in the positive end of the $Y_{\mathrm{gipm}}$-axis and meets the quasi-radial IMF occurrence rates in the negative end of the $Y_{\mathrm{gipm}}$-axis. The number of jets detected downstream of the quasi-parallel shock per hour per $R_{\mathrm{E}}$ is around 1–2, so the occurrence rates are approximately 5–10 times as high as the rates downstream of the quasi-perpendicular shock. Based on the means of the six middle bins with modest error bars, the number of jets is larger by a factor of 9 downstream of the quasi-parallel shock. Taking the error bars into account, this factor is 6–14.

In the 2D maps of Fig. 4, we present the number of jets detected per hour per $R_{\mathrm{E}}^2$ in the magnetosheath in the $X_{\mathrm{gipm}}$-$Y_{\mathrm{gipm}}$-plane. Note that the square cells in the maps are $2\,R_{\mathrm{E}} \times 2\,R_{\mathrm{E}}$ meaning that the units have been scaled from $1/(4R_{\mathrm{E}}^2)$ to $1/R_{\mathrm{E}}^2$. The white cells have $< 1,000$ MSH observations and the white cells with dashed outlines have $\geq 1,000$ MSH observations but

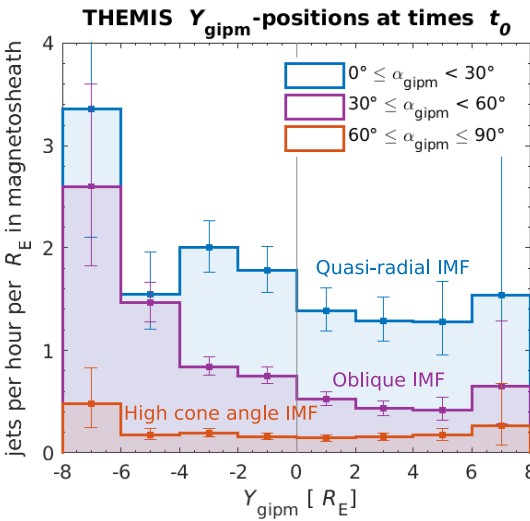

**Figure 3.** The number of jets observed per hour per $R_E$ in the subsolar magnetosheath as functions of $Y_{gipm}$ for the three cone angle ranges. The positions have been renormalized to the mean SW dynamic pressure 1.76 nPa.

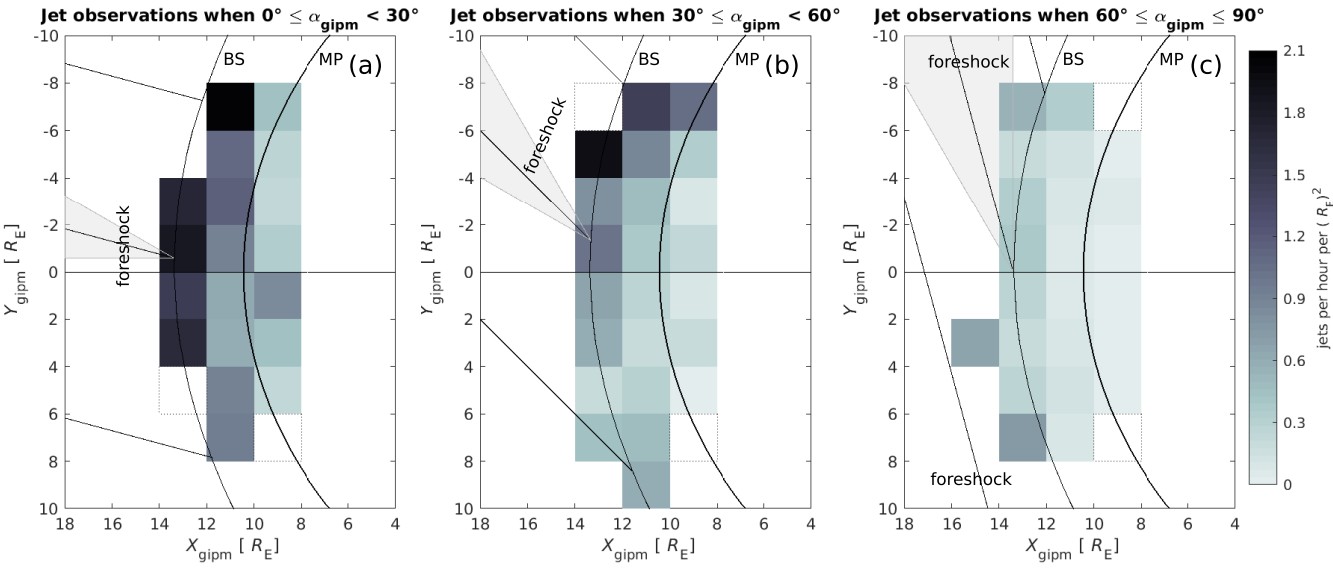

**Figure 4.** Maps showing the number of jets observed per hour per $R_E^2$ in the $X_{gipm}$-$Y_{gipm}$-plane during (**a**) quasi-radial IMF with $\alpha_{gipm} \in [0°, 30°)$, (**b**) oblique IMF with $\alpha_{gipm} \in [30°, 60°)$, and (**c**) high cone angle IMF with $\alpha_{gipm} \in [60°, 90°]$. The positions have been renormalized to the mean SW dynamic pressure 1.76 nPa. The white squares have fewer than 1,000 MSH observations. The dashed squares contain 1,000 or more MSH observations but no jets. The IMF lines on the left correspond to the middle value of the cone angle range and the cone represents the whole range of cone angles.

no jets. In addition, the maps feature magnetic field lines on the left in the solar wind representing the middle IMF cone angle value of the range. For example, in the cone angle range $[0°, 30°)$, the magnetic field lines have a cone angle of $15°$. The whole range of cone angles is represented by the gray cone.

Figure 4 shows that jets are detected more frequently close to the bow shock than close to the magnetopause as already noted by Plaschke et al. (2013). During quasi-radial IMF ($\alpha_{\text{gipm}} \in [0°, 30°)$), the jet occurrence is relatively high on the whole $Y_{\text{gipm}}$ width of the observation area. In comparison, while the occurrence rate has gone down for oblique IMF ($\alpha_{\text{gipm}} \in [30°, 60°)$), there is a strong preference for more jets occurring with decreasing $Y_{\text{gipm}}$. For higher cone angles $[60°, 90°]$, the occurrence rates are low and there is no longer a clearly visible difference between the sides $Y_{\text{gipm}} > 0$ and $Y_{\text{gipm}} < 0$. The occurrence of jets is higher on the edges of the observational area.

Comparing the results for each cone angle range to each other and looking at the distribution of jets inside each of the ranges, our results show that jets occur predominantly downstream of the expected quasi-parallel shock shown in Fig. 1 and the occurrence increases with decreasing angle $\theta_{Bn}$ between the local shock normal and the IMF. Jets do, however, also occur downstream of the quasi-perpendicular shock but less frequently. There is no clear increase in the occurrence of jets downstream of the border between the quasi-parallel and quasi-perpendicular shock regions. We would expect that such an effect would be the most easily seen for the cone angle range $[30°, 60°)$ but based on Fig. 3, there is a clear increasing trend towards negative $Y_{\text{gipm}}$.

In Figure 5, we present the estimated jet impact rates on the subsolar magnetopause reference area for three different jet sizes perpendicular to the flow direction: 0.5–1.0 $R_{\text{E}}$, 1.0–2.0 $R_{\text{E}}$ and $> 2.0$ $R_{\text{E}}$. Geoeffective jets larger than $> 2.0$ $R_{\text{E}}$ hit the magnetopause around 9.3 times per hour during quasi-radial IMF, around 4.3 times per hour during oblique IMF, and around 0.87 times per hour during high cone angle IMF. Smaller jets of perpendicular diameters 0.5–2.0 $R_{\text{E}}$ are almost constantly hitting the magnetopause: 3.3 jets per minute during quasi-radial IMF, 1.5 jets per minute during oblique IMF, and 0.31 jets per minute during high cone angle IMF. The total impact rate of all jets of these three scale sizes is 3.4 jets per minute during quasi-radial IMF and 0.33 jets per minute during high cone angle IMF.

## 4   Discussion

The data clearly supports the hypothesis that P13 jets occur more frequently downstream of the quasi-parallel region of the bow shock. The jet occurrence rate downstream of the quasi-parallel shock is 9 (6–14) times the rate downstream of the quasi-perpendicular shock in the subsolar magnetosheath. The occurrence increases as the angle between the local shock normal and the IMF gets smaller, and thus for oblique IMF there is an increasing trend of jet occurrence from the quasi-perpendicular to the quasi-parallel side. We do not see enhanced occurrence of jets downstream of the boundary between the quasi-parallel and quasi-perpendicular regions in our results. However, this effect could easily be hidden since the cone angle ranges are $30°$ wide and therefore there is presumably quite a lot of variation in the location of this boundary. Nevertheless, the effect of this boundary to jet occurrence seems to be small. We also tested performing the statistics with all jet interval data points instead of just using the time $t_0$ to represent individual jets. The results were very similar and the conclusions remained the same.

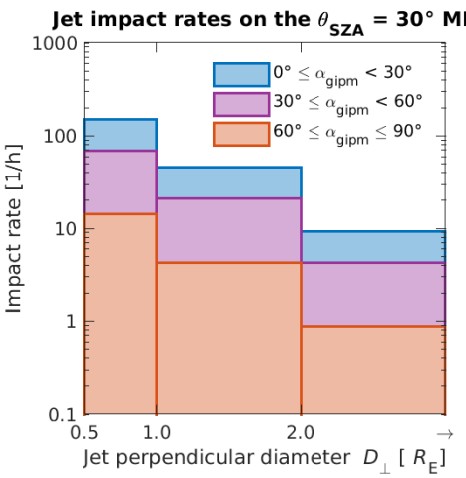

**Figure 5.** Estimated jet impact rates on the $30°$ solar zenith angle (SZA) subsolar magnetopause as a function of jet size perpendicular to the propagation direction. The estimations are based on the model introduced by Plaschke et al. (2016).

We estimated the magnetopause impact rates of jets during different IMF orientations. The most straightforward assumption would be that jets approximately follow the spatial distribution of Figure 3 when impacting the magnetopause. However, this is not necessarily true since the propagation of jets in the magnetosheath is not well known at the moment. The 2D maps of Figure 4 suggest that the general occurrence patterns in the $Y_{\mathrm{gipm}}$-direction are preserved close to the magnetopause. Moreover, according to Hietala and Plaschke (2013), the deflection angle of jet propagation direction from the magnetosheath background flow increases when moving closer to the magnetopause which indicates that jets can maintain their direction.

The main uncertainty in our study comes from the OMNI solar wind data corresponding to each of our MSH observations. The OMNI data set consist of measurements from different spacecraft around the L1 point (King and Papitashvili, 2005). The solar wind measurements have been time-shifted to the bow shock. There is uncertainty in the estimated time-shift and how solar wind structures evolve while the solar wind propagates towards Earth. There may also be local solar wind variations at the L1 point and near the Earth's bow shock. We also note that the direction of $\hat{\mathbf{Y}}_{\mathrm{gipm}}$ and therefore also the value of $Y_{\mathrm{gipm}}$ are not very well-defined when the IMF is almost parallel to the $X_{\mathrm{gipm}}$-axis, that is with the lowest IMF cone angles. For the cone angle range $[0°, 30°)$, 20 % of the jet events took place during $\alpha_{\mathrm{gipm}} < 15°$ conditions. The jet impact rate model assumes that the distribution of jet sizes is the same for all IMF orientations. However, due to jets being observed more often during low IMF cone angle conditions, the distribution of jet sizes is likely to be biased towards those jets.

The trends seen by Plaschke et al. (2013) are clear in this study: jets occur mostly during low IMF cone angle conditions and closer to the bow shock than to the magnetopause. Archer and Horbury (2013) have shown the connection between magnetosheath total dynamic pressure enhancements and the quasi-parallel shock. Our results show that this behaviour is also true for magnetosheath jets defined by Plaschke et al. (2013) that are specifically defined as enhancements of earthward dynamic

pressure that could potentially hit the magnetopause and affect the magnetosphere. Furthermore, our results are presented in a way that offers easy interpretation, quantitative numbers of jets, and direct predictability in the $\mathbf{B}_{\mathrm{IMF}}$-$\mathbf{v}_{\mathrm{SW}}$-plane.

The results suggest that foreshock processes are responsible for the generation of most jets. Suggested mechanisms supported by these results are, e.g., bow shock ripples and SLAMS which are both inherent to the quasi-parallel shock. According to Hietala and Plaschke (2013), the deflection angle of jet propagation direction from the magnetosheath background flow increases when moving closer to the magnetopause indicating that jets can maintain their earthward direction. Since jets mostly occur downstream of the quasi-parallel shock, we can therefore expect the effects of jets to be more prominent in the magnetosphere downstream of the quasi-parallel region. There are no clear deviations from this assumption close to the magnetopause in the 2D maps of Fig. 4. Hietala et al. (2018) provided evidence of a jet triggering reconnection at the magnetopause and discussed the possibility of jets being able to also suppress reconnection. Future studies will reveal whether these effects produce a non-negligible net effect on the occurrence of reconnection downstream of the quasi-parallel shock. Magnetopause surface waves, magnetopause standing waves, and auroras connected to jets colliding into the magnetopause are also expected to be excited more frequently at the magnetopause downstream of the quasi-parallel shock. The estimated magnetopause impact rates provided in this study help us to forecast these effects of jets.

## 5   Summary and Conclusions

In this study, we showed that anti-sunward jets in the subsolar magnetosheath mostly occur downstream of the quasi-parallel bow shock where the angle between the IMF and the local shock normal is small. The occurrence rates are approximately 9 times higher than downstream of the quasi-perpendicular shock. For oblique IMF the rates within the magnetosheath downstream of the bow shock follow a monotonically increasing trend from the quasi-perpendicular side towards the quasi-parallel side. This suggests that foreshock processes are responsible for jet formation with possible formation mechanisms including bow shock ripples inherent to the quasi-parallel shock and short large amplitude magnetic structures (SLAMS). However, not all jets occur downstream of the quasi-parallel shock so alternative formation mechanisms are also needed.

The occurrence pattern of magnetosheath jets presented here suggests that we can expect more of the newly found magnetopause surface eigenmodes and other jet induced phenomena to be produced at the magnetopause downstream of the quasi-parallel shock. Large jets of diameters $> 2\,R_{\mathrm{E}}$ perpendicular to the propagation direction are estimated to hit the $30°$ solar zenith angle subsolar magnetopause around 9 times in an hour during quasi-radial IMF, 4 times in an hour during oblique IMF, and once in an hour during high cone angle IMF.

*Data availability.* Data from the THEMIS mission including level 2 FGM and ESA data are publicly available from the University of California Berkeley and can be obtained from http://themis.ssl. berkeley.edu/data/themis (THEMIS, 2019). NASA's OMNI high-resolution (1 min cadence) solar wind data are also publicly available and can be obtained from ftp://spdf.gsfc. nasa.gov/pub/data/omni (NASA, 2019). The plot data of this study is available at https://doi.org/10.5281/zenodo.3333518.

*Author contributions.* HH conceived of the study. LV performed the data analysis and wrote the manuscript with contributions from HH and FP.

*Competing interests.* The authors declare that they have no conflict of interest.

*Acknowledgements.* The work in the University of Turku was supported by the Turku Collegium of Science and Medicine. The work of HH
5    was also supported by NASA grant NNX17AI45G. We acknowledge NASA contract NAS5-02099 and V. Angelopoulos for use of data from the THEMIS Mission.

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
