# Peer review of "Jets in the Magnetosheath: IMF Control of Where They Occur"

_Annales Geophysicae, 2019_

## Referee Comment (RC1) · Anonymous Referee #1 · 28 May 2019

This paper addresses the spatial distribution of magnetosheath jets as a function of the orientation of the interplanetary magnetic fields, using a combined THEMIS and OMNI data set. The paper is very well written and gives interesting results on this aspect of magnetosheath jets, which has not been studied in detail before. I have only some minor issues that I would like the authors to address or comment before publication.

1. Figure 1: How are the boundaries of the foreshock regions determined, specifically the angle of the foreshock boundary wrt the X axis?

2. page 4, lines 15-17: How exactly is the number of jets determined? For jets with a dynamic pressure marginally greater than the criterion a single jet may have a dynamic pressure that repeatedly goes above and then below this limit. are such occurrences counted as individual jets, or are they combined to one jet (similar to what is often done

for bursty bulk flow events)? If not, this may skew the statistics and overestimate the number of jets with low dynamic pressure.

3. As can be seen from Figure 2, even for low cone angles part of the subsolar region of the bow shock is associated with the quasi-perpendicular shock. It would be good to get a number of how large a part of the bow shock is quasi-perpendicular for a few cone angles.

4. page 6, line 11: 'We have used Xgipm-axis...' should read 'We have used the Xgipm-axis...'

5. page 7, line 3: 'very high error bars' should read 'very large error bars'.

6. page 7, line 9-10: 'with decreasing Ygipm, i.e. with decreasing theta_Bn'. This is not strictly true, since the angle also depends on Xgipm. Perhaps it would be instructive to plot the distributions in the 'opposite' sense as well, i.e. for a few ranges of Ygipm plot the number of jets per hour as a function of theta_Bn, although you do get a sense of this from Figure 4.

7. page 7, line 11-13: The authors mix the denotions 'quasi-radial IMF', 'quasi-parallel shock', 'high-cone angle IMF', and 'quasi-perpendicular shock'. Do you consider there to be a one-to-one correlation?

8. page 8, line 7: 'clear visible' should read 'clearly visible'.

9. page 9, line 7: 'could be easily' should read 'could easily be'.

10. page 9, line 30: 'jets are thought to be able to also suppress reconnection'. Please elaborate or give a reference.
* * *

---

## Referee Comment (RC2) · Anonymous Referee #1 · 13 Jun 2019

The authors have adressed all of my issues except part of point 2. With the definition used by the authors, it seems to me that very week jets will be overrepresented when the statistics is presented based on number of jets, rather than the number of data points that fulfill the jet criterion. I would like to see a brief discussion on this. Do you have any argument that the results would not change significantly if you used 'number of data points', instead of number of jets?

---

## Author Comment (AC1) · 13 Jun 2019

Turku, June 13, 2019

Dear Referee #1,

We thank you for taking the time to review this study and for your valuable input. Please see below a detailed response to each of your comments that are shown in italics.

1. *Figure 1: How are the boundaries of the foreshock regions determined, specifically the angle of the foreshock boundary wrt the X axis?*

Figure 1 is a sketch used to show the approximate locations of the foreshock regions. The purpose of the figure is to illustrate that the location of the foreshock region is very different for the three different cases: radial IMF, 45° cone angle IMF and 90° cone angle IMF. The foreshock regions extend upstream from the quasi-parallel shock where the particles can reflect from the shock. The quasi-parallel shock was approximately drawn as the area of the shock where the angle between the IMF and the local shock normal is less than 45°. For radial IMF in Fig. 1a, the edges of the foreshock are drawn to approximately emulate simulation results of quasi-radial IMF (e.g., Omidi et al. (2009) https://doi.org/10.1029/2008JA013950, Blanco-Cano et al. (2009) https://doi.org/10.1029/2008JA013406, and Palmroth et al. (2015) https://doi.org/10.1002/2015JA021526). In Fig. 1b and 1c, the boundaries of the foreshock regions are drawn a little bit inward (towards the Earth) from the field line that tangentially touches the bow shock because the foreshock particles drift due to the convective electric field.

If the Editor permits us to submit a revision of the manuscript, we would like to make the following changes/additions:

Page 2, line 15: "as shown" -> "as illustrated"
Figure 1 caption: "A sketch of the…"

2. *page 4, lines 15-17: How exactly is the number of jets determined? For jets with a dynamic pressure marginally greater than the criterion a single jet may have a dynamic pressure that repeatedly goes above and then below this limit. Are such occurrences counted as individual jets, or are they combined to one jet (similar to what is often done for bursty bulk flow events)? If not, this may skew the statistics and overestimate the number of jets with low dynamic pressure.*

We use the definition of magnetosheath jets described by Plaschke et al. (2013). A jet is defined such that within a jet interval the earthward dynamic pressure exceeds half of the total solar wind pressure. The whole jet interval is then defined as the time around this peak when the earthward dynamic pressure is larger than ¼ of the total solar wind dynamic pressure. Therefore, many peaks can occur within one jet interval. The jet data points used in this study are the instants of time of the maximum ratio between magnetosheath earthward dynamic pressure and total solar wind dynamic pressure within the jet interval.

If the Editor permits us to submit a revision, we would like to add to page 4 line 16: "The entire jet interval is then defined as the period when the earthward dynamic pressure is over ¼ of the total solar wind dynamic pressure."

*3. As can be seen from Figure 2, even for low cone angles part of the subsolar region of the bow shock is associated with the quasi-perpendicular shock. It would be good to get a number of how large a part of the bow shock is quasi-perpendicular for a few cone angles.*

We thank the Referee for bringing up this important question. The observation region is a 30° Earth-centered and Sun-facing cone around the $X_{gse}$-axis, which is very close to the $X_{gipm}$-axis. We can see in Figure 4 that most observations span over the $Y_{gipm}$ range of [-8 Re, 8 Re]. Looking at the model bow shocks in Figure 4, we can see that the curvature of the Earth's bow shock in this subsolar region is at most around 30°.

Let us first consider the quasi-radial IMF case (cone angles [0°,30°)). We can estimate that for 15° cone angle IMF the edge of quasi-perpendicular region is at the very bottom of Figure 4, so for cone angles [0°,15°] the whole observation area is quasi-parallel. For 30° cone angle IMF, this boundary is approximately at the bow shock point where $Y_{gipm}$ = 6 Re. Thus, almost all observations during quasi-radial IMF can be considered to be downstream of the quasi-parallel shock.

Similarly, let us consider the high cone angle IMF case (cone angles [60°,90°]). For IMF with cone angle of 60°, the edge of the quasi-parallel region is approximately at $Y_{gipm}$ = −6 Re so that most of the observation region can be considered to be downstream of the quasi-perpendicular shock. For cone angles [75°,90°], the entire observation region is estimated to be downstream of the quasi-perpendicular shock. Therefore, we can make an approximation that during high cone angle IMF our observation area is downstream of the quasi-perpendicular shock.

The oblique IMF (cone angles [30°,60°)) is the case in between. For 45° cone angle IMF, we can estimate that the positive $Y_{gipm}$ side is quasi-perpendicular and the negative $Y_{gipm}$ side is quasi-parallel.

If the Editor permits us to submit a revised version of the manuscript, we would like to add a short explanation of this to the paragraph starting on page 5 line 10.

*4. page 6, line 11: 'We have used Xgipm-axis. . .' should read 'We have used the Xgipm-axis. . .'*

If the Editor permits, we would like to correct this in the revised version in the way suggested by the Referee.

*5. page 7, line 3: 'very high error bars' should read 'very large error bars'.*

If the Editor permits, we would like to correct this in the revised version in the way suggested by the Referee.

*6. page 7, line 9-10: 'with decreasing Ygipm, i.e. with decreasing theta_Bn'. This is not strictly true, since the angle also depends on Xgipm. Perhaps it would be instructive to plot the distributions in the 'opposite' sense as well, i.e. for a few ranges of Ygipm plot the number of jets per hour as a function of theta_Bn, although you do get a sense of this from Figure 4.*

We agree with the Referee that this wording is not entirely accurate and apologize for the oversight. There is a very small dependency on $X_{gipm}$ and there is also dependency on $Z_{gipm}$ that we do not consider here.

If the Editor permits us to submit a revision, we would like to rephrase this in the form:
"with decreasing Ygipm, i.e., towards the side of the shock which is generally more quasi-parallel."

7. *page 7, line 11-13: The authors mix the denotations 'quasi-radial IMF', 'quasi-parallel shock', 'high-cone angle IMF', and 'quasi-perpendicular shock'. Do you consider there to be a one-to-one correlation?*

We thank the Referee for pointing this out. Please see our detailed analysis on this matter in point 3.

8. *page 8, line 7: 'clear visible' should read 'clearly visible'.*

If the Editor permits, we would like to correct this in the revised version in the way suggested by the Referee.

9. *page 9, line 7: 'could be easily' should read 'could easily be'.*

If the Editor permits, we would like to correct this in the revised version in the way suggested by the Referee.

10. *page 9, line 30: 'jets are thought to be able to also suppress reconnection'. Please elaborate or give a reference.*

We thank the Referee for pointing this out. Hietala et al. (2018) (https://doi.org/10.1002/2017GL076525) discussed this possibility in their paper. They suggested that the magnetic field within jets could change the magnetic shear angle at the magnetopause. Alternatively, jets could change the magnetic shear angle by indenting the magnetopause and thus perturbing the magnetospheric field lines. The changes in the magnetic shear angle could then enhance or suppress magnetic reconnection.

If the Editor permits, we would like to correct this in the revised version to be in the form: Page 9, line 29: "Hietala et al. (2018) provided evidence of a jet triggering reconnection at the magnetopause and discussed the possibility of jets also being able to suppress reconnection. Future studies will reveal whether these effects produce a non-negligible net effect on the occurrence of reconnection downstream of the quasi-parallel shock."

Should the Editor accept this manuscript for revision, we would like to add a figure showing estimations of how often jets impact the magnetopause for different IMF cone angles. These estimations have been made using the jet occurrence rates derived in this study and the formulas derived by Plaschke et al. (2016) (https://doi.org/10.1002/2016JA022534). We think the magnetopause impact rates would fit well into this paper as we discuss the predictability of jets and their effects on the magnetosphere-ionosphere system.

We would like to thank the Referee once again for their constructive comments which enabled us to improve this manuscript.

On behalf of all co-authors,
Laura Vuorinen

---

## Referee Comment (RC3) · Anonymous Referee #1 · 17 Jun 2019

The authors have addressed my remaining issue, and with the additional text they suggest (perhaps together with a comment along the lines of "This criterion prevents multiple consecutive peaks from being counted as individual jets.", as they include in the reply), I am now happy to recommend the paper for publication.
* * *

---

## Author Comment (AC2) · 17 Jun 2019

Dear Referee #1,

We thank you for the discussion. Please see our response to your comment (in italics) below:

*The authors have adressed all of my issues except part of point 2. With the definition used by the authors, it seems to me that very week jets will be overrepresented when the statistics is presented based on number of jets, rather than the number of data points that fulfill the jet criterion. I would like to see a brief discussion on this. Do you have any argument that the results would not change significantly if you used 'number of data points', instead of number of jets?*

We apologize that our previous answers were not conclusive and thank you for the interesting question. We plotted Figure 3 and Figure 4 again using all jet interval data points as you suggested. Here are the plots in comparison with the original plots shown first (left or above):

[Figure]

[Figure]

[Figure]

[Figure]

The results and trends are very similar within error bars. The ratio between the means of the six middle histogram bins (Ygipm [-6 Re, 6 Re]) of quasi-radial IMF (cone angles [0°,30°)) and high cone angle IMF ([60°,90°]) observations is again 9. Therefore, the conclusions of our study remain the same.

Although this was an excellent test, we prefer to keep using number of jets as the units of measurement instead of number of jet interval data points. Long duration jets may dominate the distribution when using all jet interval points for statistics. This happens, for example, in the new 2D plot for [30°,60°) cone angles, for the cell at the very bottom (Xgipm = [10 Re, 12 Re) and Ygipm = [8 Re, 10 Re]). This cell only contains one long duration jet.

Furthermore, the jet definition by Plaschke et al. (2013) includes a criterion that within one-minute intervals before and after the jet interval, the Xgse ion velocity in the magnetosheath has to go above half of the corresponding value at the time t_0 (the time of the highest ratio between anti-sunward dynamic pressure in the magnetosheath and in the solar wind). This criterion prevents multiple consecutive peaks from being counted as individual jets.

Should the Editor allow us to submit a revision of our manuscript, we will add the following to the Discussion section of the manuscript:
"We also tested doing the statistics with all jet interval data points instead of just using the time t_0 to represent individual jets. The results were very similar and the conclusions remained the same."

Thank you again for the discussion and for your suggestions.

On behalf of all co-authors,
Laura Vuorinen

---

## Author Comment (AC3) · 20 Jun 2019

We will add a mention about the criterion that prevents multiple consecutive peaks from being counted as individual jets into the manuscript.

We thank the Referee for their valuable input which enabled us to improve this manuscript.

On behalf of all the co-authors, Laura Vuorinen
* * *

---

## Author Response (AR1)

Turku, July 12, 2019

Dear Referee #1,

We thank you for taking the time to review this study and for your valuable input. Please see below a detailed response to each of your comments that are shown in italics.

1. *Figure 1: How are the boundaries of the foreshock regions determined, specifically the angle of the fore-shock boundary wrt the X axis?*

Figure 1 is a sketch used to show the approximate locations of the foreshock regions. The purpose of the figure is to illustrate that the location of the foreshock region is very different for the three different cases: radial IMF, 45° cone angle IMF and 90° cone angle IMF. The foreshock regions extend upstream from the quasi-parallel shock where the particles can reflect from the shock. The quasi-parallel shock was approximately drawn as the area of the shock where the angle between the IMF and the local shock normal is less than 45°. For radial IMF in Fig. 1a, the edges of the foreshock are drawn to approximately emulate simulation results of quasi-radial IMF (e.g., Omidi et al. (2009) https://doi.org/10.1029/2008JA013950, Blanco-Cano et al. (2009) https://doi.org/10.1029/2008JA013406, and Palmroth et al. (2015) https://doi.org/10.1002/2015JA021526). In Fig. 1b and 1c, the boundaries of the foreshock regions are drawn a little bit inward (towards the Earth) from the field line that tangentially touches the bow shock because the foreshock particles drift due to the convective electric field.

We have made the following changes to the manuscript:

Page 2, line 16: "as shown" -> "as illustrated"
Figure 1 caption: "A sketch of the…"

2. *page 4, lines 15-17: How exactly is the number of jets determined? For jets with a dynamic pressure marginally greater than the criterion a single jet may have a dynamic pressure that repeatedly goes above and then below this limit. Are such occurrences counted as individual jets, or are they combined to one jet (similar to what is often done for bursty bulk flow events)? If not, this may skew the statistics and overestimate the number of jets with low dynamic pressure.*

We use the definition of magnetosheath jets described by Plaschke et al. (2013). A jet is defined such that within a jet interval the earthward dynamic pressure exceeds half of the total solar wind pressure. The whole jet interval is then defined as the time around this peak when the earthward dynamic pressure is larger than ¼ of the total solar wind dynamic pressure. Therefore, many peaks can occur within one jet interval. The jet data points used in this study are the instants of time of the maximum ratio between magnetosheath earthward dynamic pressure and total solar wind dynamic pressure within the jet interval.

We have added to page 4 lines 19—20: "The entire jet interval is then defined as the period when the earthward dynamic pressure is over ¼ of the total solar wind dynamic pressure."

3. *As can be seen from Figure 2, even for low cone angles part of the subsolar region of the bow shock is associated with the quasi-perpendicular shock. It would be good to get a number of how large a part of the bow shock is quasi-perpendicular for a few cone angles.*

We thank the Referee for bringing up this important question. The observation region is a 30° Earth-centered and Sun-facing cone around the Xgse-axis, which is very close to the Xgipm-axis. We can see in Figure 4 that most observations span over the Ygipm range of [-8 Re, 8 Re]. Looking at the model bow shocks in Figure 4, we can see that the curvature of the Earth's bow shock in this subsolar region is at most around 30°.

Let us first consider the quasi-radial IMF case (cone angles [0°,30°)). We can estimate that for 15° cone angle IMF the edge of quasi-perpendicular region is at the very bottom of Figure 4, so for cone angles [0°,15°] the whole observation area is quasi-parallel. For 30° cone angle IMF, this boundary is approximately at the bow shock point where Ygipm = 6 Re. Thus, almost all observations during quasi-radial IMF can be considered to be downstream of the quasi-parallel shock.

Similarly, let us consider the high cone angle IMF case (cone angles [60°,90°]). For IMF with cone angle of 60°, the edge of the quasi-parallel region is approximately at Ygipm = −6 Re so that most of the observation region can be considered to be downstream of the quasi-perpendicular shock. For cone angles [75°,90°], the entire observation region is estimated to be downstream of the quasi-perpendicular shock. Therefore, we can make an approximation that during high cone angle IMF our observation area is downstream of the quasi-perpendicular shock.

The oblique IMF (cone angles [30°,60°)) is the case in between. For 45° cone angle IMF, we can estimate that the positive Ygipm side is quasi-perpendicular and the negative Ygipm side is quasi-parallel.

We have made changes on page 5 lines 12—13: "quasi-radial IMF when almost all of the dayside magnetosheath observations can be considered to be downstream of the quasi-parallel shock", and on lines 14—15: "high cone angle IMF when all of the dayside magnetosheath observations can be considered to be downstream of the quasi-perpendicular shock".

4. *page 6, line 11: 'We have used Xgipm-axis. . .' should read 'We have used the Xgipm-axis. . .'*

We have corrected this in the way suggested by the Referee on page 6 line 13.

5. *page 7, line 3: 'very high error bars' should read 'very large error bars'.*

We have corrected this in the way suggested by the Referee on page 7 line 16.

6. *page 7, line 9-10: 'with decreasing Ygipm, i.e. with decreasing theta_Bn'. This is not strictly true, since the angle also depends on Xgipm. Perhaps it would be instructive to plot the distributions in the 'opposite' sense as well, i.e. for a few ranges of Ygipm plot the number of jets per hour as a function of theta_Bn, although you do get a sense of this from Figure 4.*

We agree with the Referee that this wording is not entirely accurate and apologize for the oversight. There is a very small dependency on Xgipm and there is also dependency on Zgipm that we do not consider here.

We have changed this on page 7 line 23 to the form: "with decreasing Ygipm, i.e., towards the side of the shock which is generally more quasi-parallel."

*7. page 7, line 11-13: The authors mix the denotations 'quasi-radial IMF', 'quasi-parallel shock', 'high-cone angle IMF', and 'quasi-perpendicular shock'. Do you consider there to be a one-to-one correlation?*

We thank the Referee for pointing this out. Please see our detailed analysis on this matter in point 3.

*8. page 8, line 7: 'clear visible' should read 'clearly visible'.*

We have corrected this in the way suggested by the Referee on page 9 line 8.

*9. page 9, line 7: 'could be easily' should read 'could easily be'.*

We have corrected this in the way suggested by the Referee on page 9 line 30.

*10. page 9, line 30: 'jets are thought to be able to also suppress reconnection'. Please elaborate or give a reference.*

We thank the Referee for pointing this out. Hietala et al. (2018) (https://doi.org/10.1002/2017GL076525) discussed this possibility in their paper. They suggested that the magnetic field within jets could change the magnetic shear angle at the magnetopause. Alternatively, jets could change the magnetic shear angle by indenting the magnetopause and thus perturbing the magnetospheric field lines. The changes in the magnetic shear angle could then enhance or suppress magnetic reconnection.

We have changed this on page 11 lines 10—14 to the form: "Hietala et al. (2018) provided evidence of a jet triggering reconnection at the magnetopause and discussed the possibility of jets also being able to suppress reconnection. Future studies will reveal whether these effects produce a non-negligible net effect on the occurrence of reconnection downstream of the quasi-parallel shock."

*11. The authors have adressed all of my issues except part of point 2. With the definition used by the authors, it seems to me that very week jets will be overrepresented when the statistics is presented based on number of jets, rather than the number of data points that fulfill the jet criterion. I would like to see a brief discussion on this. Do you have any argument that the results would not change significantly if you used 'number of data points', instead of number of jets?*

We apologize that our previous answers were not conclusive and thank you for the interesting question. We plotted Figure 3 and Figure 4 again using all jet interval data points as you suggested. Here are the plots in comparison with the original plots shown first (left or above):

[Figure]

The results and trends are very similar within error bars. The ratio between the means of the six middle histogram bins (Ygipm [-6 Re, 6 Re]) of quasi-radial IMF (cone angles [0°,30°)) and high cone angle IMF ([60°,90°]) observations is again 9. Therefore, the conclusions of our study remain the same.

Although this was an excellent test, we prefer to keep using number of jets as the units of measurement instead of number of jet interval data points. Long duration jets may dominate the distribution when using all jet interval points for statistics. This happens, for example, in the new 2D plot for [30°,60°) cone angles, for the cell at the very bottom (Xgipm = [10 Re, 12 Re) and Ygipm = [8 Re, 10 Re]). This cell only contains one long duration jet.

Furthermore, the jet definition by Plaschke et al. (2013) includes a criterion that within one-minute intervals before and after the jet interval, the Xgse ion velocity in the magnetosheath has to go above half of the corresponding value at the time t_0 (the time of the highest ratio between anti-sunward dynamic pressure in the magnetosheath and in the solar wind). This criterion prevents multiple consecutive peaks from being counted as individual jets.

We made the following addition on page 9 line 32: "We also tested performed the statistics with all jet interval data points instead of just using the time t_0 to represent individual jets. The results were very similar and the conclusions remained the same."

12. *The authors have addressed my remaining issue, and with the additional text they suggest (perhaps together with a comment along the lines of "This criterion prevents multiple consecutive peaks from being counted as individual jets.", as they include in the reply), I am now happy to recommend the paper for publication.*

We have added a mention of this criterion on page 4 lines 21—23: "In order to prevent multiple consecutive peaks from being counted as individual jets, within the one-minute long windows before and after the jet interval, the Xgse ion velocity has to go above half of the corresponding value at t_0."

We have added estimations of how often jets impact the magnetopause for different IMF cone angles using the jet occurrence rates derived in this study and the formulas derived by Plaschke et al. (2016) (https://doi.org/10.1002/2016JA022534). We explain the methods we used in Section 2.6 on page 7. The results are shown in Figure 5 on page 10. The new version of the plot data of this study is now found at https://doi.org/10.5281/zenodo.3333518.

Thank you again for the helpful comments which allowed us to improve this manuscript.

On behalf of all co-authors,
Laura Vuorinen

[revised manuscript text omitted]